# First Molecular Confirmation of *Treponema* spp. in Lesions Consistent with Digital Dermatitis in Chilean Dairy Cattle

**DOI:** 10.3390/pathogens11050510

**Published:** 2022-04-26

**Authors:** Nivia Canales, Hedie Bustamante, Jennifer Wilson-Welder, Cristian Thomas, Emilio Ramirez, Miguel Salgado

**Affiliations:** 1Laboratorio de Enfermedades Infecciosas, Instituto de Medicina Preventiva Veterinaria, Facultad de Ciencias Veterinarias, Universidad Austral de Chile, Casilla, P.O. Box 567, Valdivia 5090000, Chile; niviaca.mor@gmail.com (N.C.); cristianfthomasp@gmail.com (C.T.); emilioramirezmv@gmail.com (E.R.); 2Escuela de Graduados, P.O. Box 567, Valdivia 5090000, Chile; 3Institute de Ciencias Clínicas Veterinarias, Facultad de Ciencias Veterinarias, Universidad Austral de Chile, Casilla, P.O. Box 567, Valdivia 5090000, Chile; hbustamante@uach.cl; 4Infectious Bacterial Diseases of Livestock Research Unit, National Animal Disease Center, Agricultural Research Service, United States Department of Agriculture, Ames, IA 50010, USA; jennifer.wilson-welder@usda.gov

**Keywords:** digital dermatitis, *Treponema* spp., PCR, dairy cattle

## Abstract

Digital dermatitis (DD) is a highly contagious and infectious disease in cattle which has a considerable negative economic impact worldwide, and adversely affects animal welfare. Members of the genus *Treponema* are the only bacterial agents for which there is consistent evidence of participation in DD lesions. In Chile, DD has been described since the 1990s, but only under a clinical approach. To date, the presence of the pathogenic agent has not been confirmed in Chile by any type of confirmatory microbiological diagnosis. The aim of the present study was to detect the presence of *Treponema* spp. DNA in lesions consistent with DD, in Chilean dairy cattle for the first time. We provide PCR confirmation of *Treponema* spp. in Chilean dairy cattle affected by DD. The high rate of positive results, as well as the proportion of the main *Treponema* species involved, is in line with what have been described in published studies elsewhere. Future herd control plans should benefit from the molecular detection of these pathogenic bacteria associated with DD.

## 1. Introduction

Digital dermatitis (DD) is a highly contagious and infectious disease that mainly affects dairy cows, negatively impacting these animals’ welfare. Moreover, DD-associated lameness and pain cause a severe reduction in milk yield, and increased culling rates [1,2]. DD has become an increasing problem in beef cattle, and recent reports of DD in other species (sheep, goats, wild elk, Mediterranean Buffalo and European Bison) demonstrates the broad infectivity potential of DD pathogens [1,3,4,5]. This important infectious disease of cattle is considered to be widespread throughout the world [6,7,8]. However, in Chile, it has only been described clinically [9,10]. 

DD is defined as a polybacterial disease that affects ruminants, clinically characterized by a painful ulcerative lesion along the coronary band, mainly affecting the area located between the heel bulbs [11,12,13]. It is considered one of the main causes of infectious lameness in domestic ruminants, especially in cattle [11]. Members of the *Treponema* genus are the only bacterial agents consistently found in DD lesions [11,14], and the fact that these pathogens thrive in humid, unhygienic or high-density housing conditions [15] makes prevention of transmission and infection difficult. This prevention must be based on perfect hygienic circumstances, no purchasing cattle, and immediate treatment of the infected ones. Furthermore, DD is characterized by limited response to herd level or systemically applied antibiotic treatments, which makes adequate treatment a challenging endeavor [16,17]. 

Microbiologically, several *Treponema* spp. are represented in DD lesions. These fastidious anaerobic spirochetes are difficult to grow, and have similar growth requirement and characteristics to each other. It is difficult to distinguish one from another and to obtain pure clonal isolates [2,18]. For this reason, during the last few decades, diagnosis of DD and detection of these spirochetes has been performed using molecular detection approaches, mainly the conventional Polymerase Chain Reaction (PCR). 

Understanding the importance of the microbiological confirmation of infectious diseases [19], PCR delivers a sensitive, specific and rapid method for detection [20,21]. Since bacteriological diagnostic confirmation of DD was still pending in Chile, we aimed to confirm the presence of *Treponema* spp. in lesions with clinical presentation consistent with DD for the first time in Chilean dairy cattle, in order to further develop future herd control plans.

## 2. Results

*Treponema* spp. were detected in 100% of the biopsies obtained from lesions with clinical presentation of DD. The most abundant *Treponema* species was *T. medium,* with 25 positive results (93%), followed by *T. phagedenis,* which was detected in 17 (63%) samples. *T. pedis* which was detected in only 6 (22%) samples and the *T. denticola* was present in only 1 (3.7%) biopsy sample (Table 1). Five samples were positive for three species (*T. phagedenis, T. medium*, and *T. pedis*) (Figure 1). Sixteen samples were positive for two species (14 for *T. phagedenis* and *T. medium*, 1 for *T. medium* and *T. pedis* and 1 for *T. denticola* and *T. pedis*). Eight samples were positive for a single species, *T. medium*. 

## 3. Discussion

The present study confirmed the presence of *Treponema* spp. in 100 % of the sampled cows with clinical presentation of lesions consistent with DD and represents the first molecular diagnosis of DD in Chilean dairy cattle. The majority of the samples (70%) were positive for two or more *Treponema* spp., and eight samples were positive for a single species, typically associated with DD.

Lameness is the second most serious condition that affects the health of dairy cattle [22]. Furthermore, lameness is a visual sign of animal welfare [23]. Increasingly, consumers are demanding higher levels of animal welfare, trackability and accountability for food production animals [23]. Depending on the geographic region, data suggests that 10–40% of all cases of lameness can be attributed specifically to digital dermatitis (DD) [24]. 

Previously, only the clinical presentation and gross description of DD had been reported in Chilean dairy cows [9,10]. Although clinical diagnosis is the most commonly used diagnostic procedure in Chile to describe DD, its use has been associated with some technical drawbacks, including the need for a high level of clinical specialization, its increasing subjectivity, and limited diagnostic sensitivity. Therefore, the diagnosis of infections such as DD, which are associated with fastidious bacteria, such as *Treponema* spp., has benefited greatly from molecular detection. Non-culture-based molecular testing has been shown to have the advantage of avoiding the days’ or weeks’ wait—compared to conventional culture—and to allow early recognition and treatment as an animal health imperative. The combined use of the 16S rRNA gene and the 16S rDNA gene for some *Treponema* phylotypes’ genetic markers assured few or low false positive results [19,25]. 

The presence of the four main *Treponema* species associated with DD in cattle [2,15,26,27] was confirmed in Chilean dairy cows. The frequencies and proportions of each of the *Treponema* species detected are similar but not the same with those reported by Mamuad et al. (2020) and by Beninger et al. (2018), who found that *T. phagedenis* and *T. medium* were the most abundant species in DD in dairy farms in Alberta and Saskatchewan, as well as in one abattoir between Calgary and Ponoka, Canada. On the other hand, Moreira et al. (2018) described that *T. phagedenis* and *T. pedis* were the most abundant species in DD cases in Brazil [2].

In addition to the detection of *Treponema* species in these lesions, the proportions at which they were found provide interesting infectological information. There is a relationship between macroscopic morphological changes, or extension of DD lesions, and the composition of the *Treponema* species and their abundance [27,28]. Another interesting aspect that has been highlighted in the pathogenesis of DD is the description that *Treponema* species detected in greater proportions, are also those that penetrate deeper into the affected tissue [2,29]. This finding has been corroborated in more severe cases which generate a more painful lesion that has a greater impact on milk production level and welfare of the infected and affected animal. According to what was observed in the field during sampling, lesions indicative of DD was clinically characterized by severe epidermal damage and epithelial proliferation, exudation and erosion. However, the precise etiopathogenesis of such lesions is not yet clear; it is apparent that *T. medium* and *T. phagedenis* were the major pathogenic bacteria detected in lesions of the bovine hoof sampled in the present study. Further study is needed to evaluate the competition, synergism, and individual pathogenic contributions (i.e., proteases, lipases, or invasion factors) of these *Treponema* spp. and their role in DD lesions. 

This first molecular confirmation of DD in dairy cattle in Chile should apprise us of the progression of this infection, insomuch as both the infection rate detected and the proportion of the main species involved are not different from that described in studies published elsewhere [27,28,30]. In addition, a bacteriological approach to DD may play a pivotal role in the management of infected cows and facilitate any control strategy in the herd.

The control of DD should be similar to that described for other infectious diseases affecting dairy cattle. Nevertheless, in Chile, several factors may be associated with an increased risk of developing DD in the future. Some of these include, among others, the limited knowledge by dairy producers of this disease and its consequences, the currently low number of diagnoses performed by bovine practitioners, and the absence of standardized husbandry practices that would limit DD infection. These factors, along with the nonexistence of proper epidemiological studies determining herd prevalence of DD put Chile at an increasing risk for the spreading of this infectious disease among and within dairy farms.

## 4. Materials and Methods

### 4.1. Study Design

A targeted convenience sampling strategy was used to maximize the probability of testing cattle infected with *Treponema* spp. by selecting herds with a history of lameness, and clinical cases consistent with DD.

### 4.2. Study Population

The study was carried out on 16 dairy herds located in the Los Ríos and Los Lagos regions, Chile, between January and May 2021. The herds were representative of the most common dairy farms in the study area in terms of breed (Holstein), size of herd (200–500 animals) and management practices (direct pasture grazing throughout the year, milked twice a day, average milk production, 220,000–4,500,000 L/year). All herds were self-reported to be closed animal populations. According to information provided by the farmers and their vets, the selected herds showed a DD clinical diagnostic rate between 5% and 20% (Table 1). Verbal consent was obtained from all producers who participated in the study.

### 4.3. Collection of Samples

A cross-sectional study was conducted, and a convenient sample of 27 biopsies obtained from lesions consistent with DD, all active lesions scored degree 3 or 4 based on the Iowa DD Lesion Scoring System [11], were collected during the study period. On average, the obtained biopsies ranged between 1 and 2 per farm on trim day. The totality of the samples was obtained by a veterinary specialist in podopathology. Typical DD lesions were identified, and the area was cleaned prior to lesion biopsy sampling. Biopsies were performed under local anesthesia using a sterile 4-5 mm skin biopsy punch and suspended in PBS 1x. All lesions biopsies were taken under local anesthesia, using 2% Lidocaine, and received proper treatment and were bandaged immediately after the biopsy procedure. The sampling was carried out in strict accordance with the Universidad Austral de Chile’s Guide for the Use of Animals for Research (www.uach.cl/direccion/investigacion/uso_animales.htm accessed on 5 January 2021). The samples were immediately transported to the Laboratory of Infectious Diseases, Instituto de Medicina Preventiva Veterinaria, Universidad Austral de Chile, to be stored at −80 °C until further processing.

### 4.4. Bacteriological Analysis of Lesions Biopsies

#### 4.4.1. Extraction of DNA

Biopsies were thawed at room temperature and DNA extraction was performed using the PureLink Genomic DNA Mini Kit (Thermo Fisher Scientific, Waltham, MA, USA), according to the manufacturer’s instructions for mammalian tissue. The concentration and quality of the DNA was evaluated in the UV-Vis NanoDrop Lite spectrophotometer (Thermo Fischer Scientific, Waltham, MA, USA). Purified DNA was stored at −20 °C until analysis.

#### 4.4.2. Molecular Confirmation

A conventional nested PCR analysis was used to detect and classify *Treponema* spp., according to Evans et al. [19,20]. Amplification was carried out using the Labnet Optimax Thermal Cycler, with initial denaturation at 94 °C for 5 min, followed by 25 cycles at 94 °C for 1 min, 55 °C for 3 min, and an extension at 72 °C for 3 min, with a final extension at 72 °C for 7 min. The amplified fragment was separated by electrophoresis on a 1.5% agarose gel and visualized under UV light on a gel documentation system. All the templates of the positive samples from the first PCR were exposed to a second PCR where 1 μL of the initial PCR was used as a template, but specific primers for the 16S rDNA gene for the *Treponema* genus and *Treponema* species (*Treponema phagedenis*, *Treponema medium*, *Treponema pedis*, and *Treponema denticola*) were included [19,25]. The conditions for the second PCR were similar to those for the initial PCR, except for the alignment temperature, which differed from one primer to another. As positive controls, genomic DNA of each of the respective phylotypes was used (kindly provided by the Infectious Bacterial Disease Research Unit, Agricultural Research Service, USDA National Animal Disease Center, Ames, IA, USA). As a negative control, mix PCR and sterile water were used. The products of the PCR were visualized by electrophoresis on a 1.5% agarose gel stained with SYBR Safe DNA Gel Stain (Invitrogen, Waltham, MA, USA).

#### 4.4.3. Descriptive Data Analysis

Venn diagram of *Treponeme* species identified by sample created using online tool Venny 2.1 (Oliveros 2007–2015) Venny. An interactive tool for comparing lists with Venn’s diagrams). https://bioinfogp.cnb.csic.es/tools/venny/index.html accessed on 2 February 2022).

## Figures and Tables

**Figure 1 pathogens-11-00510-f001:**
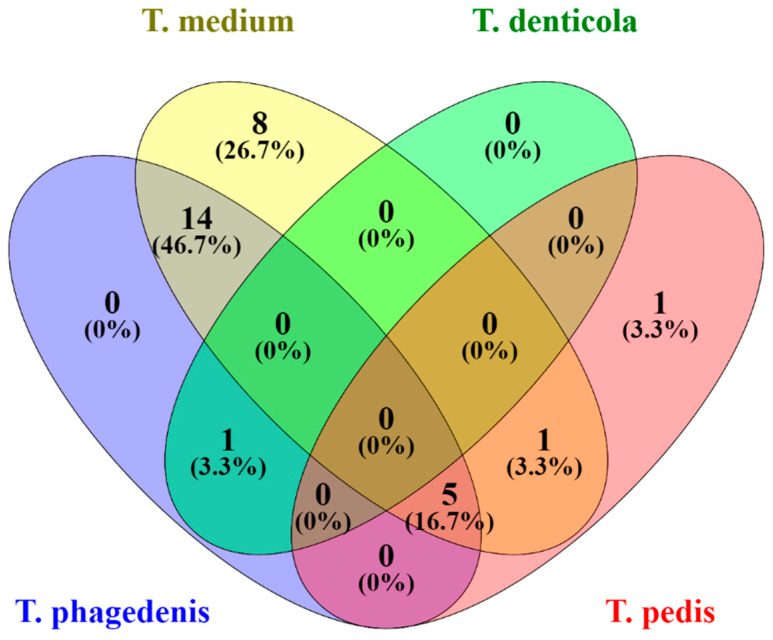
Venn diagram of samples illustrating number (and percentage) of samples that were positive for each *Treponema* species. Depth of shading corresponds to percentage of total samples.

**Table 1 pathogens-11-00510-t001:** *Treponema* spp. detected by PCR in affected DD tissue of cows in southern Chile.

ID	Farm	Lameness Prev. %	DD Prev. %	Treponema	T. Phagedenis	T. Medium	T. Denticola	T. Pedis
31	7	20	10	+	+	+	−	+
582	2			+	+	+	−	−
1033	11	10	7	+	+	+	−	+
1359	14	8	5	+	+	+	−	+
1746	1	15	15	+	+	+	−	−
1784	10	7	6	+	+	+	−	−
2152	7	20	10	+	+	+	−	+
222-7986	1	15	15	+	+	+	−	−
3065	10	7	6	+	+	−	+	−
3310	6			+	+	+	−	−
4213	14	8	5	+	+	+	−	−
4258	13	7	6	+	+	+	−	−
4352	7	20	10	+	+	+	−	−
4552	7	20	10	+	+	+	−	−
5415	8	7	6	+	+	+	−	−
5841	14	8	5	+	−	+	−	−
5877	14	8	5	+	−	+	−	−
7871	12	5	5	+	+	+	−	−
8865	5	15	15	+	−	+	−	−
9235	3	7	6	+	−	+	−	−
943-6669	14	8	5	+	+	+	−	−
9456	3	7	6	+	−	+	−	+
9505	7	20	10	+	−	+	−	−
9508	7	20	10	+	−	+	−	−
9542	4			+	−	+	−	−
9585	7	20	10	+	−	+	−	−
9766	14	8	5	+	−	−	−	+
**Percentage**				**100%**	**63%**	**93%**	**3%**	**22%**

DD: Digital Dermatitis, +: Positive result, −: Negative result.

## Data Availability

Not applicable.

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
