# Peer review of "First Molecular Confirmation of Treponema spp. in Lesions Consistent with Digital Dermatitis in Chilean Dairy Cattle"

_pathogens, 2022, doi:10.3390/pathogens11050510_

Round 1

Reviewer 1 Report

The authors described the Treponema spp. in DD lesions in Chilean dairy cattle. The topic is interesting and well written. However, I have a few questions:

1) It was unclear how you obtained the convenient sample of 27 animals from 16 herds. Did you start a whole herd screening? Or did you know which cows were more likely to be affected by DD based on the records and went to exam those cows? It would be good to know the proportions of the different DD stages using IOWA system. It would be better to include the "prevalence" figure and discuss its difference and similarity to the pasture-based dairy system in other countries with proper references. 

2) It would be better to describe the IOWA system with a bit more details and cite the proper references.

3) Are there any new info regarding DD prevalence in Chile, the source of info does not have to be the published articles. It would be interesting to read this info.

4) Statistical inferences are required if you want to estimate some population attributes based on the samples. Based on the design, the study showed four treponema species were present on Chilean dairy farms, but it was not able to describe the distribution of the four treponema species in Chilean, therefore I do not think including hypothesis testing is of any use. It might be misleading. Suggest to delete this part. Descriptive stats would be sufficient.

Minor errors:

L163 Bacteriological analysis. Then what?

L194 Ven or Venn?

There are many old references cited in the paper, some of which could be replaced by more recent publications. This shows the authors have up-to-date knowledge of the disease.

Author Response

Reviewer 1:

The authors described the Treponema spp. in DD lesions in Chilean dairy cattle. The topic is interesting and well written. However, I have a few questions:

  • It was unclear how you obtained the convenient sample of 27 animals from 16 herds. Did you start a whole herd screening? Or did you know which cows were more likely to be affected by DD based on the records and went to exam those cows? It would be good to know the proportions of the different DD stages using IOWA system. It would be better to include the "prevalence" figure and discuss its difference and similarity to the pasture-based dairy system in other countries with proper references. 

Answer: The main objective of the study was the molecular detection of members of the Treponema genus in lesions consistent with digital dermatitis for the first time in Chile. For this reason, convenience sampling was peformed at trim day for cows with obvious active lesions. We hope that in future studies we can collect epidemiological information such as that suggested by the reviewer.

  • It would be better to describe the IOWA system with a bit more details and cite the proper references.

Answer: consistent with the previous answer. The Iowa system and its equivalent of the M system were used only to make sure collection of samples from active lesions. It was not the objective of the present study to analyze the categories.

  • Are there any new info regarding DD prevalence in Chile, the source of info does not have to be the published articles. It would be interesting to read this info.

Answer: in references 9 and 10 some prevalence information is given. As for example, Rodriguez et al (1998) point out that the prevalence of digital dermatitis in the central Regions of Chile, between 1995 to 1998 fluctuated in 6.1% to 10%. As a group, we hope to update this type of epidemiological information in futures studies.

  • Statistical inferences are required if you want to estimate some population attributes based on the samples. Based on the design, the study showed four treponema species were present on Chilean dairy farms, but it was not able to describe the distribution of the four treponema species in Chilean, therefore I do not think including hypothesis testing is of any use. It might be misleading. Suggest to delete this part. Descriptive stats would be sufficient.

Answer: we agree with the reviewer. Therefore, we will proceed as suggested. See Descriptive data analysis (page 6 line 195 new version) and Results (page 2 line 59 new version).

Minor errors:

  • L163 Bacteriological analysis. Then what?

Answer: the subheading has been supplemented to improve the presentation of activities related to bacteriological analysis. See page 5, line 169 of new version.

  • L194 Ven or Venn?

Answer: must be Venn. It has been corrected as suggested. See page 2, line 68 new version.

  • There are many old references cited in the paper, some of which could be replaced by more recent publications. This shows the authors have up-to-date knowledge of the disease.

Answer: we humbly disagree with the reviewer, since we consider rather appropriate the cited references. The references of Rodriguez et al, is the oldest (1998), but it represents significant information in the area of this infectious disease in Chile.

There are references from 2004 (nº 6 and 7), but these are the first reports from some countries such as Turkey and New Zealand.

Then come references from 2006. The reference 16 talks about antibiotic treatments and the 21 that talks about the application of molecular biology to infectious diseases.

The rest of the references have been published less than 10 years ago. More specifically, 8 out of 30 references cited, are less than 5 years old.

Reviewer 2 Report

Dear authors,

Thank you for your valuble information about the confirmation of diffrent Treponema spp. in DD lesions. I have advised the Journal to give you the opportunity to improve the manuscript, so that it makes it a more valuble contribution. At this moment it gives too much space for questions.

page 1, line 18: in how many herds and in how many cattle

line 30: do these references give information about decline in reproduction?

line 42: I would state here that prevention of transmission and infection must be based on: perfect hygienic circumstances, no purchasing cattle, immediately treatment of infected  etc.

page 2, line 50: abbreviations like PCR should be write out for the first time that this is used.

Results

I am not familiar with your region and as I have understood from M&M that cows are pastured 24 h/day, what are the risk moments of transmission? And I missed here the information about average herd size, parity and DIM of cows sampled and was ditribution of the different species equal over different parities and DIM. We had no acces to the diagram.

page 3, Table 1. you have results of 27 samples out of 16 herds, so some herds provide > 1, with a max of 2?

Discussion

line 77: for two spp., in 5 herds also T denticola and in only 1 herd T. pedis was approved

next part is repeatment of introduction and can be skipped

line 97-103: it seemed that in other studies different Trep. spp. were found, that is interesting and reason for discussion

line 113: is there any indication which Trep. spp. were responsible?

M&M

line 145: we know from different other studies that this is most times an underestimation

line 152: on avreage 2 means that you still can have sampled in one herd 9 cows??

line 151: not all readers of this Journal are familiar with that scoring system, so we need more info here

Moleculair confirmation gives CT -values, that provides the reader information about how strong the positivity was

Looking forward to receive your improved manuscript 

Author Response

Reviewer 2:

Comments and Suggestions for Authors

Dear authors,

Thank you for your valuble information about the confirmation of diffrent Treponema spp. in DD lesions. I have advised the Journal to give you the opportunity to improve the manuscript, so that it makes it a more valuble contribution. At this moment it gives too much space for questions.

  • page 1, line 18: in how many herds and in how many cattle,

Answer: this information is contained in the reference: Rodriguez-Lainz, A., Melendez-Retamal, P., Hird, D. W., and Read, D. H. Papillomatous digital dermatitis in Chilean dairies and evaluation of a screening method. Preventive Veterinary Medicine, 1998, 37(1–4), 197–207. The study population consisted of 214 dairy farms, with almost 30,000 cattle.

  • line 30: do these references give information about decline in reproduction?

Answer: the reviewer is correct. Therefore, to be more precise in relation to the content of the references, reproductive information has been deleted from the sentence. See page 1 line 30 of new version.

  • line 42: I would state here that prevention of transmission and infection must be based on: perfect hygienic circumstances, no purchasing cattle, immediately treatment of infected  etc.,

Answer: a new sentence has been included, as suggested. See pages 1 and 2, lines 42-44 of new version.

  • page 2, line 50: abbreviations like PCR should be write out for the first time that this is used.,

Answer: corrected as suggested. Page 2, line 52 new version.

Results

  • I am not familiar with your region and as I have understood from M&M that cows are pastured 24 h/day, what are the risk moments of transmission? And I missed here the information about average herd size, parity and DIM of cows sampled and was ditribution of the different species equal over different parities and DIM. We had no acces to the diagram.

Answer: this type of epidemiological information was not investigated, since it was not the aim of the present study. We only focused on trying to detect and report for the first time in Chile the presence of Treponema species from active lesions of digital dermatitis in dairy cattle. We hope that future studies that we can do will provide details of this.

  • page 3, Table 1. you have results of 27 samples out of 16 herds, so some herds provide > 1, with a max of 2?

Answer: it is correct what the reviewer reports, on average were 1 or 2 animals per herd.

Discussion

  • line 77: for two spp., in 5 herds also T denticola and in only 1 herd T. pedis was approved

next part is repeatment of introduction and can be skipped.

Answer: corrected as suggested. See Page 3 lines 80-81 new version.

  • line 97-103: it seemed that in other studies different Trep. spp. were found, that is interesting and reason for discussion

Answer: the sentence has been modified for clarity. See page 4 lines 102-106 new version.

  • line 113: is there any indication which Trep. spp. were responsible?

Answer: the required information has been added, as suggested. See page 4 lines 119-120 new version.

M&M

  • line 145: we know from different other studies that this is most times an underestimation.

Answer: we agree with the reviewer that the information provided by both the farmer and his veterinary adviser may be underestimated. The first thing to do in order to solve this conflict is to confirm that the infection exists in their herds, then describe the species of Treponema involved and we hope that future studies will help to generate and specify the true rate of infection in Chile.

  • line 152: on avreage 2 means that you still can have sampled in one herd 9 cows??

Answer: the reviewer is correct. The sentence has been modified for clarity. See page 5, line 158 new version.

  • line 151: not all readers of this Journal are familiar with that scoring system, so we need more info here

Answer: due to the fact that room in the short communication format is an essential requirement, we prefer to provide the respective citation, so that the reader can be enlightened in relation to the lesion DD scoring system followed. See page 5, line 157 new version.

  • Moleculair confirmation gives CT -values, that provides the reader information about how strong the positivity was, que dice?

Answer: a conventional nested PCR analysis was used to detect and classify Treponema spp, and this PCR system does not inform a positive reaction with CT values, as in a qPCR system.

Looking forward to receive your improved manuscript 

We really appreciate your comments and inputs that without doubt enrich the new version of our manuscript

Reviewer 3 Report

The authors described for the first time the molecular identification of Treponema spp. in digital dermatitis (DD) lesion in dairy cattle. DD is a multifactorial disease, and multiple bacteria are associated with the disease, being Treponema spp. consistently detected in the lesion. However, to date there is no confirmation of DD etiopathogenesis, and it should be reflected and discussed in this manuscript.

This study does not contain any sample from healthy animals (from infected or non-infected farms) or environmental samples from farms without DD cases, if sampling healthy animals is a major concern. This samples would allow to further relate the presence of Treponema spp. and DD. Also it would allow to know the prevalence of this spirochetes in a environment or country not tested before. Therefore, although the first identification of Treponema spp. in DD lesion is interesting the lack of sampling healthy animals or DD free farms environment limit the usefulness of the current study. This point should at least be mentioned in the discussion.

In addition, the presence of different lesion grades and the correlation of Treponema spp. is discussed in the discussion and data are not provided in the results.  I would like to suggest to the author to include the lesion degree or score or description and its relation to the detected species. It would result in an interesting description of the observed cases.   

Figure 1 is missing, at least in my document.

L18. Microbiological confirmation for diagnosis should be done by culture and no with molecular methods, therefore I suggest to remove “of confirmatory” from the L19.

L24: Consider to replace “molecular confirmation” by “molecular detection”

L24. Consider replacement of and “of this serious infectious disease in cattle” for “of these pathogenic bacteria associated with DD”.

L34-35 Please, rephrase.

L488-55: Very long sentence difficult to follow. Please consider to split in several sentences.

L81. Please include a reference.

L91. Consider to replace “are caused” for “are associated”.

L96. Consider to replace “No false positive results” for “few or low false positive results”.

L110-114 Consider to include the data in the results.  

Author Response

Reviewer 3:

Comments and Suggestions for Authors

  • The authors described for the first time the molecular identification of Treponema spp. in digital dermatitis (DD) lesion in dairy cattle. DD is a multifactorial disease, and multiple bacteria are associated with the disease, being Treponema spp.consistently detected in the lesion. However, to date there is no confirmation of DD etiopathogenesis, and it should be reflected and discussed in this manuscript.

Answer: this information has been included in the Discussion part, as suggested. See page 4 lines 118-121 new version.

  • This study does not contain any sample from healthy animals (from infected or non-infected farms) or environmental samples from farms without DD cases, if sampling healthy animals is a major concern. This samples would allow to further relate the presence of Treponema spp.and DD. Also it would allow to know the prevalence of this spirochetes in a environment or country not tested before. Therefore, although the first identification of Treponema in DD lesion is interesting the lack of sampling healthy animals or DD free farms environment limit the usefulness of the current study. This point should at least be mentioned in the discussion.

Answer: we agree with the reviewer that these design criteria should be considered in future studies; however, the objective of our study was to confirm the presence of treponeme in lesions consistent wit DD. Therefore, extra epidemiological information, although highly necessary, it is not in the scope of the present study.

  • In addition, the presence of different lesion grades and the correlation of Treponema spp.is discussed in the discussion and data are not provided in the results.  I would like to suggest to the author to include the lesion degree or score or description and its relation to the detected species. It would result in an interesting description of the observed cases.

Answer: we agree, it is important information, however, we basically work with samples stage 2 or 3 of the Iowa system and M2 in the M grade system so that the study is aimed at animals with active disease. Future studies of our group are already considered the analysis of different lesion degree with the presence and number of Treponema spp.

  • Figure 1 is missing, at least in my document.

Answer: added as suggested. See page 2, line 68 new version.

  • Microbiological confirmation for diagnosis should be done by culture and no with molecular methods, therefore I suggest to remove “of confirmatory” from the L19.

Answer: corrected as suggested. See page 1, lines 19 and 20 new version

  • L24: Consider to replace “molecular confirmation” by “molecular detection”

Answer: corrected as suggested. See page 1, line 24 new version

  • Consider replacement of and “of this serious infectious disease in cattle” for “of these pathogenic bacteria associated with DD”.

Answer: corrected as suggested. See page 1, line 24 new version

  • L34-35 Please, rephrase.

Answer: the sentence has been rephrased, as suggested. See page 1, lines 33-34 new version.

  • L48-55: Very long sentence difficult to follow. Please consider to split in several sentences.

Answer: the paragraph has been splitted in several sentence, as suggested. See page 2 lines 47-58 new version.

  • Please include a reference.

Answer: the reference has been included, as suggested. See page 3, line 83 new version.

  • Consider to replace “are caused” for “are associated”.

Answer: corrected as suggested. See page 4, line 93 new version

  • Consider to replace “No false positive results” for “few or low false positive results”.

Answer: corrected as suggested. See page 4, line 98 new version

  • L110-114 Consider to include the data in the results.

Answer: new information has been included, as suggested. See p4, lines 118-121 new version.

Round 2

Reviewer 1 Report

I do not think the authors addressed most of my concerns.

Author Response

Answer: we show you again our responses to the reviewer's last comments:

  • It was unclear how you obtained the convenient sample of 27 animals from 16 herds. Did you start a whole herd screening? Or did you know which cows were more likely to be affected by DD based on the records and went to exam those cows? It would be good to know the proportions of the different DD stages using IOWA system. It would be better to include the "prevalence" figure and discuss its difference and similarity to the pasture-based dairy system in other countries with proper references. 

Answer: The main objective of the study was the molecular detection of members of the Treponema genus in lesions consistent with digital dermatitis for the first time in Chile. For this reason, convenience sampling was peformed at trim day for cows with obvious active lesions. We hope that in future studies we can collect epidemiological information such as that suggested by the reviewer.

  • It would be better to describe the IOWA system with a bit more details and cite the proper references.

Answer: consistent with the previous answer. The Iowa system and its equivalent of the M system were used only to make sure collection of samples from active lesions. It was not the objective of the present study to analyze the categories.

  • Are there any new info regarding DD prevalence in Chile, the source of info does not have to be the published articles. It would be interesting to read this info.

Answer: in references 9 and 10 some prevalence information is given. As for example, Rodriguez et al (1998) point out that the prevalence of digital dermatitis in the central Regions of Chile, between 1995 to 1998 fluctuated in 6.1% to 10%. As a group, we hope to update this type of epidemiological information in futures studies.

  • Statistical inferences are required if you want to estimate some population attributes based on the samples. Based on the design, the study showed four treponema species were present on Chilean dairy farms, but it was not able to describe the distribution of the four treponema species in Chilean, therefore I do not think including hypothesis testing is of any use. It might be misleading. Suggest to delete this part. Descriptive stats would be sufficient.

Answer: we agree with the reviewer. Therefore, we will proceed as suggested. See Descriptive data analysis (page 6 line 195 new version) and Results (page 2 line 59 new version).

Minor errors:

  • L163 Bacteriological analysis. Then what?

Answer: the subheading has been supplemented to improve the presentation of activities related to bacteriological analysis. See page 5, line 169 of new version.

  • L194 Ven or Venn?

Answer: must be Venn. It has been corrected as suggested. See page 2, line 68 new version.

  • There are many old references cited in the paper, some of which could be replaced by more recent publications. This shows the authors have up-to-date knowledge of the disease.

Answer: we humbly disagree with the reviewer, since we consider rather appropriate the cited references. The references of Rodriguez et al, is the oldest (1998), but it represents significant information in the area of this infectious disease in Chile.

There are references from 2004 (nº 6 and 7), but these are the first reports from some countries such as Turkey and New Zealand.

Then come references from 2006. The reference 16 talks about antibiotic treatments and the 21 that talks about the application of molecular biology to infectious diseases.

The rest of the references have been published less than 10 years ago. More specifically, 8 out of 30 references cited, are less than 5 years old.

Reviewer 2 Report

Dear authors,

in my opinion you have improved the manuscript seriously, but I still have some questions:

page 1, line 65: is not different is not correct in my opinion. I would prefer "is in line with..."

page 5, line 194: decreased animal welfare?

line 229: not clear yet, 

page 6, Collection of samples, how was the are cleaned with acohol 70% or water or ?? 

Author Response

page 1, line 65: is not different is not correct in my opinion. I would prefer "is in line with..."

Answer: Although the page and line number coordinates of the present edit relative to the latest version of the manuscript is not entirely accurate, the sentence has been corrected as suggested. See page (P) 1, line (L) 22 of the new version (NV).

Page 5, line 194: decreased animal welfare?

Answer: Unfortunately, the page and line number coordinates of the present edit relative to the latest version of the manuscript are not clear, so it is not possible for us to respond appropriately to the reviewer's concern

line 229: not clear yet, 

Answer: Unfortunately, the page and line number coordinates of the present edit relative to the latest version of the manuscript are not clear, so it is not possible for us to respond appropriately to the reviewer's concern

page 6, Collection of samples, how was the are cleaned with acohol 70% or water or ?? 

Answer: yes, the area was washed with water, dried with absorbent paper, and then disinfected with 70º alcohol. Subsequently, the biopsy was obtained as described in the text.

Reviewer 3 Report

The author address all the comments appropriately.

Author Response

Answer: we really appreciate the valuable contribution of reviewer 3.